# Error-Minimizing Estimates and Universal Entry-Wise Error Bounds for Low-Rank Matrix Completion

**Franz J. Király**[*]
Department of Statistical Science and
Centre for Inverse Problems
University College London
f.kiraly@ucl.ac.uk

**Louis Theran**[†]
Institute of Mathematics
Discrete Geometry Group
Freie Universität Berlin
theran@math.fu-berlin.de

## Abstract

We propose a general framework for reconstructing and denoising single entries of incomplete and noisy entries. We describe: effective algorithms for deciding if and entry can be reconstructed and, if so, for reconstructing and denoising it; and a priori bounds on the error of each entry, individually. In the noiseless case our algorithm is exact. For rank-one matrices, the new algorithm is fast, admits a highly-parallel implementation, and produces an error minimizing estimate that is qualitatively close to our theoretical and the state-of-the-art Nuclear Norm and OptSpace methods.

## 1 Introduction

Matrix Completion is the task to reconstruct low-rank matrices from a subset of its entries and occurs naturally in many practically relevant problems, such as missing feature imputation, multi-task learning [2], transductive learning [4], or collaborative filtering and link prediction [1, 8, 9]. Almost all known methods performing matrix completion are optimization methods such as the max-norm and nuclear norm heuristics [3, 9, 10], or OptSpace [5], to name a few amongst many. These methods have in common that in general: (a) they reconstruct the whole matrix; (b) error bounds are given for all of the matrix, not single entries; (c) theoretical guarantees are given based on the sampling distribution of the observations. These properties are all problematic in scenarios where: (i) one is interested only in predicting or imputing a specific set of entries; (ii) the entire data set is unwieldy to work with; (iii) or there are non-random "holes" in the observations. All of these possibilities are very natural for the typical "big data" setup.

The recent results of [6] suggest that a method capable of handling challenges (i)–(iii) is within reach. By analyzing the algebraic-combinatorial structure Matrix Completion, the authors provide algorithms that identify, for *any* fixed set of observations, *exactly* the entries that can be, in principle, reconstructed from them. Moreover, the theory developed indicates that, when a missing entry can be determined, it can be found by first exposing combinatorially-determined polynomial relations between the known entries and the unknown ones and then selecting a common solution.

To bridge the gap between the theory of [6] and practice are the following challenges: to efficiently find the relevant polynomial relations; and to extend the methodology to the noisy case. In this paper, we show how to do both of these things in the case of rank one, and discuss how to instantiate the same scheme for general rank. It will turn out that finding the right set of polynomials and

---

[*]Supported by the Mathematisches Forschungsinstitut Oberwolfach

[†]Supported by the European Research Council under the European Union's Seventh Framework Programme (FP7/2007-2013) / ERC grant agreement no 247029-SDModels.

noisy estimation are intimately related: we can treat each polynomial as providing an estimate of the missing entry, and we can then take as our estimate the variance minimizing weighted average. This technique also gives *a priori lower bounds* for a broad class of unbiased single-entry estimators in terms of the *combinatorial structure* of the observations and the *noise model* only. In detail, our contributions include:

- the construction of a variance-minimal and unbiased estimator for any fixed missing entry of a rank-one-matrix, under the assumption of known noise variances

- an explicit form for the variance of that estimator, being a lower bound for the variance of any unbiased estimation of any fixed missing entry and thus yielding a quantiative measure on the trustability of that entry reconstructed from any algorithm

- the description of a strategy to generalize the above to any rank

- comparison of the estimator with two state-of-the-art optimization algorithms (OptSpace and nuclear norm), and error assessment of the three matrix completion methods with the variance bound

As mentioned, the restriction to rank one is not inherent in the overall scheme. We depend on rank one only in the sense that we understand the combinatorial-algebraic structure of rank-one-matrix completion exactly, whereas the behavior in higher rank is not yet as well understood. Nonetheless, it is, in principle accessible, and, once available will can be "plugged in" to the results here without changing the complexity much. In this sense, the present paper is a proof-of-concept for a new approach to estimating and denoising in algebraic settings based on *combinatorially enumerating* a set of *polynomial estimators* and then averaging them. For us, computational efficiency comes via a connection to the topology of graphs that is specific to this problem, but we suspect that this part, too, can be generalized somewhat.

## 2 The Algebraic Combinatorics of Matrix Completion

We first briefly review facts about Matrix Completion that we require. The exposition is along the lines of [6].

**Definition 2.1.** *A matrix $M \in \{0,1\}^{m \times n}$ is called a mask. If $A$ is a partially known matrix, then the mask of $A$ is the mask which has ones in exactly the positions which are known in $A$ and zeros otherwise.*

**Definition 2.2.** *Let $M$ be an $(m \times n)$ mask. We will call the unique bipartite graph $G(M)$ which has $M$ as bipartite adjacency matrix the completion graph of $M$. We will refer to the $m$ vertices of $G(M)$ corresponding to the rows of $M$ as blue vertices, and to the $n$ vertices of $G(M)$ corresponding to the columns as red vertices. If $e = (i, j)$ is an edge in $K_{m,n}$ (where $K_{m,n}$ is the complete bipartite graph with $m$ blue and $n$ red vertices), we will also write $A_e$ instead of $A_{ij}$ and for any $(m \times n)$ matrix $A$.*

A fundamental result, [6, Theorem 2.3.5], says that identifiability and reconstructability are, up to a null set, graph properties.

**Theorem 2.3.** *Let $A$ be a generic[1] and partially known $(m \times n)$ matrix of rank $r$, let $M$ be the mask of $A$, let $i, j$ be integers. Whether $A_{ij}$ is reconstructible (uniquely, or up to finite choice) depends only on $M$ and the true rank $r$; in particular, it does not depend on the true $A$.*

For rank one, as opposed to higher rank, the set of reconstructible entries is easily obtainable from $G(M)$ by combinatorial means:

**Theorem 2.4** ([6, Theorem 2.5.36 (i)]). *Let $G \subseteq K_{m,n}$ be the completion graph of a partially known $(m \times n)$ matrix $A$. Then the set of uniquely reconstructible entries of $A$ is exactly the set $A_e$, with $e$ in the transitive closure of $G$. In particular, all of $A$ is reconstructible if and only if $G$ is connected.*

## 2.1 Reconstruction on the transitive closure

We extend Theorem 2.4's theoretical reconstruction guarantee by describing an explicit, algebraic algorithm for actually doing the reconstruction.

**Definition 2.5.** *Let $P \subseteq K_{m,n}$ (or, $C \subseteq K_{m,n}$) be a path (or, cycle), with a fixed start and end. We will denote by $E^+(P)$ be the set of edges in $P$ (resp. $E^+(C)$ and $C$) traversed from blue vertex to a red one, and by $E^-(P)$ the set of edges traversed from a red vertex to a blue one [2]. From now on, when we speak of "oriented paths" or "oriented cycles", we mean with this sign convention and some fixed traversal order.*

*Let $A = A_{ij}$ be a $(m \times n)$ matrix of rank 1, and identify the entries $A_{ij}$ with the edges of $K_{m,n}$. For an oriented cycle $C$, we define the polynomials*

$$P_C(A) = \prod_{e \in E^+(C)} A_e - \prod_{e \in E^-(C)} A_e, \quad and$$

$$L_C(A) = \sum_{e \in E^+(C)} \log A_e - \sum_{e \in E^-(C)} \log A_e,$$

*where for negative entries of $A$, we fix a branch of the complex logarithm.*

**Theorem 2.6.** *Let $A = A_{ij}$ be a generic $(m \times n)$ matrix of rank 1. Let $C \subseteq K_{m,n}$ be an oriented cycle. Then, $P_C(A) = L_C(A) = 0$.*

*Proof:* The determinantal ideal of rank one is a binomial ideal generated by the $(2 \times 2)$ minors of $A$ (where entries of $A$ are considered as variables). The minor equations are exactly $P_C(A)$, where $C$ is an elementary oriented four-cycle; if $C$ is an elementary 4-cycle, denote its edges by $a(C)$, $b(C)$, $c(C)$, $d(C)$, with $E^+(C) = \{a(C), d(C)\}$. Let $\mathcal{C}$ be the collection of the elementary 4-cycles, and define $L_{\mathcal{C}}(A) = \{L_C(A) : C \in \mathcal{C}\}$ and $P_{\mathcal{C}}(A) = \{P_C(A) : C \in \mathcal{C}\}$.

By sending the term $\log A_e$ to a formal variable $x_e$, we see that the free $\mathbb{Z}$-group generated by the $L_C(A)$ is isomorphic to $H_1(K_{m,n}, \mathbb{Z})$. With this equivalence, it is straightforward that, for any oriented cycle $D$, $L_D(A)$ lies in the $\mathbb{Z}$-span of elements of $L_{\mathcal{C}}(A)$ and, therefore, formally,

$$L_D(A) = \sum_{C \in \mathcal{C}} \alpha_C \cdot L_C(A)$$

with the $\alpha_C \in \mathbb{Z}$. Thus $L_D(\cdot)$ vanishes when $A$ is rank one, since the r.h.s. does. Exponentiating completes the proof. $\qquad \square$

**Corollary 2.7.** *Let $A = A_{ij}$ be a $(m \times n)$ matrix of rank 1. Let $v, w$ be two vertices in $K_{m,n}$. Let $P, Q$ be two oriented paths in $K_{m,n}$ starting at $v$ and ending at $w$. Then, for all $A$, it holds that $L_P(A) = L_Q(A)$.*

# 3 A Combinatorial Algebraic Estimate for Missing Entries and Their Error

We now construct our estimator.

## 3.1 The sampling model

In all of the following, we will assume that the observations arise from the following sampling process:

**Assumption 3.1.** *There is an unknown fixed, rank one, matrix $A$ which is generic, and an $(m \times n)$ mask $M \in \{0, 1\}^{m \times n}$ which is known. There is a (stochastic) noise matrix $\mathcal{E} \in \mathbb{R}^{m \times n}$ whose entries are uncorrelated and which is multiplicatively centered with finite variance, non-zero[3] variance; i.e., $\mathbb{E}(\log \mathcal{E}_{ij}) = 0$ and $0 < \mathrm{Var}(\log \mathcal{E}_{ij}) < \infty$ for all $i$ and $j$.*

*The observed data is the matrix $A \circ M \circ \mathcal{E} = \Omega(A \circ \mathcal{E})$, where $\circ$ denotes the Hadamard (i.e., component-wise) product. That is, the observation is a matrix with entries $A_{ij} \cdot M_{ij} \cdot \mathcal{E}_{ij}$.*

The assumption of multiplicative noise is a necessary precaution in order for the presented estimator (and in fact, any estimator) for the missing entries to have bounded variance, as shown in Example 3.2 below. This is not, in practice, a restriction since an infinitesimal additive error $\delta A_{ij}$ on an entry of $A$ is equivalent to an infinitesimal multiplicative error $\delta \log A_{ij} = \delta A_{ij}/A_{ij}$, and additive variances can be directly translated into multiplicative variances if the density function for the noise is known[4]. The previous observation implies that the multiplicative noise model is as powerful as any additive one that allows bounded variance estimates.

**Example 3.2.** *Consider a $(2 \times 2)$-matrix $A$ of rank 1. The unique equation between the entries is then $A_{11}A_{22} = A_{12}A_{21}$. Solving for any entry will have another entry in the denominator, for example $A_{11} = \frac{A_{12}A_{21}}{A_{22}}$. Thus we get an estimator for $A_{11}$ when substituting observed and noisy entries for $A_{12}, A_{21}, A_{22}$. When $A_{22}$ approaches zero, the estimation error for $A_{11}$ approaches infinity. In particular, if the density function of the error $E_{22}$ of $A_{22}$ is too dense around the value $-A_{22}$, then the estimate for $A_{11}$ given by the equation will have unbounded variance. In such a case, one can show that no estimator for $A_{11}$ has bounded variance.*

## 3.2 Estimating entries and error bounds

In this section, we construct the unbiased estimator for the entries of a rank-one-matrix with minimal variance. First, we define some notation to ease the exposition:

**Notations 3.3.** *We will denote by $a_{ij} = \log A_{ij}$ and $\varepsilon_{ij} = \log \mathcal{E}_{ij}$ the logarithmic entries and noise. Thus, for some path $P$ in $K_{m,n}$ we obtain*

$$L_P(A) = \sum_{e \in E^+(P)} a_e - \sum_{e \in E^-(P)} a_e.$$

*Denote by $b_{ij} = a_{ij} + \varepsilon_{ij}$ the logarithmic (observed) entries, and $B$ the (incomplete) matrix which has the (observed) $b_{ij}$ as entries. Denote by $\sigma_{ij} = \mathrm{Var}(b_{ij}) = \mathrm{Var}(\varepsilon_{ij})$.*

The components of the estimator will be built from the $L_P$:

**Lemma 3.4.** *Let $G = G(M)$ be the graph of the mask $M$. Let $x = (v, w) \in K_{m,n}$ be any edge with $v$ red. Let $P$ be an oriented path in $G(M)$ starting at $v$ and ending at $w$. Then,*

$$L_P(B) = \sum_{e \in E^+(P)} b_e - \sum_{e \in E^-(P)} b_e$$

*is an unbiased estimator for $a_x$ with variance $\mathrm{Var}(L_P(B)) = \sum_{e \in P} \sigma_e$.*

*Proof:* By linearity of expectation and centeredness of $\varepsilon_{ij}$, it follows that

$$\mathbb{E}(L_P(B)) = \sum_{e \in E^+(P)} \mathbb{E}(b_e) - \sum_{e \in E^-(P)} \mathbb{E}(b_e),$$

thus $L_P(B)$ is unbiased. Since the $\varepsilon_e$ are uncorrelated, the $b_e$ also are; thus, by Bienaymé's formula, we obtain

$$\mathrm{Var}(L_P(B)) = \sum_{e \in E^+(P)} \mathrm{Var}(b_e) + \sum_{e \in E^-(P)} \mathrm{Var}(b_e),$$

and the statement follows from the definition of $\sigma_e$.

In the following, we will consider the following parametric estimator as a candidate for estimating $a_e$:

**Notations 3.5.** *Fix an edge $x = (v, w) \in K_{m,n}$. Let $\mathcal{P}$ be a basis for the $v$–$w$ path space and denote $\#\mathcal{P}$ by $p$. For $\alpha \in \mathbb{R}^p$, set $X(\alpha) = \sum_{P \in \mathcal{P}} \alpha_P L_P(B)$.*

*Furthermore, we will denote by $\mathbb{1}$ the $n$-vector of ones.*

The following Lemma follows immediately from Lemma 3.4 and Theorem 2.6:

**Lemma 3.6.** $\mathbb{E}(X(\alpha)) = \mathbb{1}^\top \alpha \cdot b_x$; *in particular, $X(\alpha)$ is an unbiased estimator for $b_x$ if and only if $\mathbb{1}^\top \alpha = 1$.*

We will now show that minimizing the variance of $X(\alpha)$ can be formulated as a quadratic program with coefficients entirely determined by $a_x$, the measurements $b_e$ and the graph $G(M)$. In particular, we will expose an explicit formula for the $\alpha$ minimizing the variance. The formula will make use of the following *path kernel*. For fixed vertices $s$ and $t$, an $s$–$t$ path is the sum of a cycle $H_1(G, \mathbb{Z})$ and $-a_{st}$. The $s$–$t$ *path space* is the linear span of all the $s$–$t$ paths. We discuss its relevant properties in Appendix A.

**Definition 3.7.** *Let $e \in K_{m,n}$ be an edge. For an edge $e$ and a path $P$, set $c_{e,P} = \pm 1$ if $e \in E^\pm(P)$ otherwise $c_{e,P} = 0$. Let $P, Q \in \mathcal{P}$ be any fixed oriented paths. Define the (weighted) path kernel $k : \mathcal{P} \times \mathcal{P} \to \mathbb{R}$ by*

$$k(P, Q) = \sum_{e \in K_{m,n}} c_{e,P} \cdot c_{e,Q} \cdot \sigma_e.$$

Under our assumption that $\mathrm{Var}(b_e) > 0$ for all $e \in K_{m,n}$, the path kernel is positive definite, since it is a sum of $p$ independent positive semi-definite functions; in particular, its kernel matrix has full rank. Here is the variance-minimizing unbiased estimator:

**Proposition 3.8.** *Let $x = (s, t)$ be a pair of vertices, and $\mathcal{P}$ a basis for the $s$–$t$ path space in $G$ with $p$ elements. Let $\Sigma$ be the $p \times p$ kernel matrix of the path kernel with respect to the basis $\mathcal{P}$. For any $\alpha \in \mathbb{R}^p$, it holds that $\mathrm{Var}(X(\alpha)) = \alpha^\top \Sigma \alpha$. Moreover, under the condition $\mathbb{1}^\top \alpha = 1$, the variance $\mathrm{Var}(X(\alpha))$ is minimized by $\alpha = \left(\Sigma^{-1} \mathbb{1}\right) \left(\mathbb{1}^\top \Sigma^{-1} \mathbb{1}\right)^{-1}.$*

*Proof:* By inserting definitions, we obtain

$$X(\alpha) = \sum_{P \in \mathcal{P}} \alpha_P L_P(B) = \sum_{P \in \mathcal{P}} \alpha_P \sum_{e \in K_{m,n}} c_{e,P} b_e.$$

Writing $b = (b_e) \in \mathbb{R}^{mn}$ as vectors, and $C = (c_{e,P}) \in \mathbb{R}^{p \times mn}$ as matrices, we obtain $X(\alpha) = b^\top C \alpha$. By using that $\mathrm{Var}(\lambda \cdot) = \lambda^2 \mathrm{Var}(\cdot)$ for any scalar $\lambda$, and independence of the $b_e$, a calculation yields $\mathrm{Var}(X(\alpha)) = \alpha^\top \Sigma \alpha$. In order to determine the minimum of the variance in $\alpha$, consider the Lagrangian

$$L(\alpha, \lambda) = \alpha^\top \Sigma \alpha + \lambda \left(1 - \sum_{P \in \mathcal{P}} \alpha_P\right),$$

where the slack term models the condition $\mathbb{1}^\top \alpha = 1$. An straightforward computation yields

$$\frac{\partial L}{\partial \alpha} = 2\Sigma \alpha - \lambda \mathbb{1}$$

Due to positive definiteness of $\Sigma$ the function $\mathrm{Var}(X(\alpha))$ is convex, thus $\alpha = \Sigma^{-1} \mathbb{1} / \mathbb{1}^\top \Sigma^{-1} \mathbb{1}$ will be the unique $\alpha$ minimizing the variance while satisfying $\mathbb{1}^\top \alpha = 1$. □

**Remark 3.9.** *The above setup works in wider generality: (i) if $\mathrm{Var}(b_e) = 0$ is allowed and there is an $s$–$t$ path of all zero variance edges, the path kernel becomes positive semi-definite; (ii) similarly if $\mathcal{P}$ is replaced with any set of paths at all, the same may occur. In both cases, we may replace $\Sigma^{-1}$ with the Moore-Penrose pseudo-inverse and the proposition still holds: (i) reduces to the exact reconstruction case of Theorem 2.4; (ii) produces the optimal estimator with respect to $\mathcal{P}$, which is optimal provided that $\mathcal{P}$ is spanning, and adding paths to $\mathcal{P}$ does not make the estimate worse.*

Our estimator is optimal over a fairly large class.

**Theorem 3.10.** *Let $\widehat{A}_{ij}$ be any estimator for an entry $A_{ij}$ of the true matrix that is: (i) unbiased; (ii) a deterministic piecewise smooth function of the observations; (iii) independent of the noise model. Let $A_{ij}^*$ be the estimator from Proposition 3.8. Then $\mathrm{Var}(A_{ij}^*) \leq \mathrm{Var}(\widehat{A}_{ij})$.*

We give a complete proof in the full version. Here, we prove the special case of log-normal noise, which gives an alternate viewpoint on the path kernel.

*Proof:* As above, we work with the formal logarithm $a_{ij}$ of $A_{ij}$. For log-normal noise, the $\varepsilon_e$ are independently distributed normals with variance $\sigma_e$. It then follows that, for any $P$ in the $i$–$j$ path space,

$$L_P(B) \sim N\left(a_{ij}, \sum_{e \in P} \sigma_e\right)$$

and the kernel matrix $\Sigma$ of the path kernel is the covariance matrix for the $L_P$ in our path basis. Thus, the $L_P$ have distribution $N(a_{ij}\mathbb{1}, \Sigma)$. It is well-known that any multivariate normal has a linear repreameterization so that the coordinates are independent; a computation shows that, here, $\Sigma^{-1}\mathbb{1}\left(\mathbb{1}^\top \Sigma^{-1}\mathbb{1}\right)^{-1}$ is the correct linear map. Thus, the estimator $A_{ij}^*$ is the sample mean of the coordinates in the new parameterization. Since this is a sufficient statistic, we are done via the Lehmann–Scheffé Theorem. $\qquad\square$

### 3.3 Rank $2$ and higher

An estimator for rank 2 and higher, together with a variance analysis, can be constructed similarly once all the solving polynomials are known. The main difficulties lies in the fact that these polynomials are not parameterized by cycles anymore, but specific subgraphs of $G(M)$, see [6, Section 2.5] and that they are not necessarily linear in the missing entry $A_e$. However, even with approximate oracles for evaluating these polynomials and estimating their covariances, an estimator similar to $X(\alpha)$ can be constructed and analyzed; in particular, we still need only to consider a basis for the space of "circuits" through the missing entry and not a costly brute force enumeration.

### 3.4 The algorithms

We now give the algorithms for estimating/denoising entries and computing the variance bounds; an implementation is available from [7]. Since the the path matrix $C$, the path kernel matrix $\Sigma$, and the optimal $\alpha$ are required for both, we show how to compute them first. We can find a basis

---

**Algorithm 1** Calculates path kernel $\Sigma$ and $\alpha$.
*Input:* index $(i, j)$, an $(m \times n)$ mask $M$, variances $\sigma$.
*Output:* path matrix $C$, path kernel $\Sigma$ and minimizer $\alpha$.

1: Find a linearly independent set of paths $\mathcal{P}$ in the graph $G(M)$, starting from $i$ and ending at $j$.
2: Determine the matrix $C = (c_{e,P})$ with $e \in G(M), P \in \mathcal{P}$; set $c_{e,P} = \pm 1$ if $e \in E^\pm(P)$, otherwise $c_{e,P} = 0$.
3: Define a diagonal matrix $S = \mathrm{diag}(\sigma)$, with $S_{ee} = \sigma_e$ for $e \in G(M)$.
4: Compute the kernel matrix $\Sigma = C^\top SC$.
5: Calculate $\alpha = \left(\Sigma^{-1}\mathbb{1}\right)\left(\mathbb{1}^\top \Sigma^{-1}\mathbb{1}\right)^{-1}$.
6: Output $C, \Sigma$ and $\alpha$.

---

for the path space in linear time. To keep the notation manageable, we will conflate formal sums of the $x_e$, cycles in $H_1(G, \mathbb{Z})$ and their representations as vectors in $\mathbb{R}^m$. Correctness is proven in Appendix A.

---

**Algorithm 2** Calculates a basis $\mathcal{P}$ of the path space.
*Input:* index $(i, j)$, an $(m \times n)$ mask $M$.
*Output:* a basis $\mathcal{P}$ for the space of oriented $i$–$j$ paths.

1: If $(i, j)$ is not an edge of $M$, and $i$ and $j$ are in different connected components, then $\mathcal{P}$ is empty. Output $\emptyset$.
2: Otherwise, if $(i, j)$ is not an edge, of $M$, add a "dummy" copy.
3: Compute a spanning forest $F$ of $M$ that does not contain $(i, j)$, if possible.
4: For each edge $e \in M \setminus F$, compute the fundamental cycle $C_e$ of $e$ in $F$.
5: If $(i, j)$ is an edge in $M$, output $\{-x_{(i,j)}\} \cup \{C_e - x_{(i,j)} : e \in M \setminus F\}$.
6: Otherwise, let $P_{(i,j)} = C_{(i,j)} - x_{(i,j)}$. Output $\{C_e - P_{(i,j)} : e \in M \setminus (F \cup \{(i,j)\})\}$.

---

Algorithms 3 and 4 then can make use of the calculated $C, \alpha, \Sigma$ to determine an estimate for any entry $A_{ij}$ and its minimum variance bound. The algorithms follow the exposition in Section 3.2, from where correctness follows; Algorithm 3 additionally provides treatment for the sign of the entries.

---

**Algorithm 3** Estimates the entry $a_{ij}$.
*Input:* index $(i, j)$, an $(m \times n)$ mask $M$, log-variances $\sigma$, the partially observed and noisy matrix $B$.
*Output:* The variance-minimizing estimate for $A_{ij}$.

---

1: Calculate $C$ and $\alpha$ with Algorithm 1.
2: Store $B$ as a vector $b = (\log |B_e|)$ and a sign vector $s = (\operatorname{sgn} B_e)$ with $e \in G(M)$.
3: Calculate $\widehat{A}_{ij} = \pm \exp\left(b^\top C \alpha\right)$. The sign is $+$ if each column of $s^\top |C|$ ($|.|$ component-wise) contains an odd number of entries $-1$, else $-$.
4: Return $\widehat{A}_{ij}$.

---

**Algorithm 4** Determines the variance of the entry $\log(A_{ij})$.
*Input:* index $(i, j)$, an $(m \times n)$ mask $M$, log-variances $\sigma$.
*Output:* The variance lower bound for $\log(A_{ij})$.

---

1: Calculate $\Sigma$ and $\alpha$ with Algorithm 1.
2: Return $\alpha^\top \Sigma \alpha$.

---

Algorithm 4 can be used to obtain the variance bound independently of the observations. The variance bound is relative, due to its multiplicativity, and can be used to approximate absolute bounds when any (in particular not necessarily the one from Algorithm 3) reconstruction estimate $\widehat{A}_{ij}$ is available. Namely, if $\widehat{\sigma}_{ij}$ is the estimated variance of the logarithm, we obtain an upper confidence/deviation bound $\widehat{A}_{ij} \cdot \exp\left(\sqrt{\widehat{\sigma}_{ij}}\right)$ for $\widehat{A}_{ij}$, and a lower confidence/deviation bound $\widehat{A}_{ij} \cdot \exp\left(-\sqrt{\widehat{\sigma}_{ij}}\right)$, corresponding to the log-confidence $\log \widehat{A}_{ij} \pm \sqrt{\widehat{\sigma}_{ij}}$. Also note that if $A_{ij}$ is not reconstructible from the mask $M$, then the deviation bounds will be infinite.

## 4 Experiments

### 4.1 Universal error estimates

For three different masks, we calculated the predicted minimum variance for each entry of the mask. The mask sizes are all $140 \times 140$. The multiplicative noise was assumed to be $\sigma_e = 1$ for each entry. Figure 1 shows the predicted a-priori minimum variances for each of the masks. The structure of the mask affects the expected error. Known entries generally have least variance, and it is less than the initial variance of 1, which implies that the (independent) estimates coming from other paths can be used to successfully denoise observed data. For unknown entries, the structure of the mask is mirrored in the pattern of the predicted errors; a diffuse mask gives a similar error on each missing entry, while the more structured masks have structured error which is determined by combinatorial properties of the completion graph.

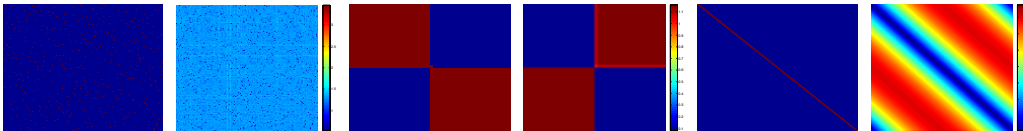

Figure 1: The figure shows three pairs of masks and predicted variances. A pair consists of two adjacent squares. The left half is the mask which is depicted by red/blue heatmap with red entries known and blue unknown. The right half is a multicolor heatmap with color scale, showing the predicted variance of the completion. Variances were calculated by our implementation of Algorithm 4.

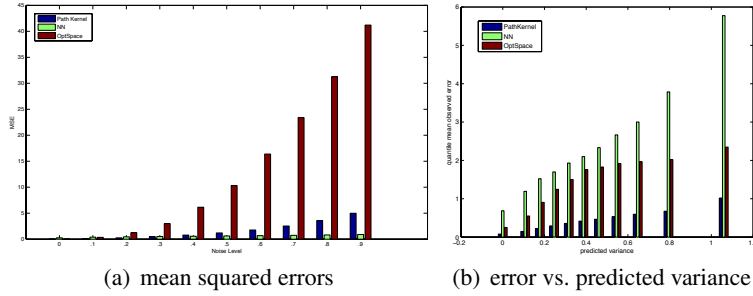

(a) mean squared errors      (b) error vs. predicted variance

Figure 2: For 10 randomly chosen masks and $50 \times 50$ true matrix, matrix completions were performed with Nuclear Norm (green), OptSpace (red), and Algorithm 3 (blue) under multiplicative noise with variance increasing in increments of $0.1$. For each completed entry, minimum variances were predicted by Algorithm 4. 2(a) shows the mean squared error of the three algorithms for each noise level, coded by the algorithms' respective colors. 2(b) shows a bin-plot of errors (y-axis) versus predicted variances (x-axis) for each of the three algorithms: for each completed entry, a pair (predicted error, true error) was calculated, predicted error being the predicted variance, and the actual prediction error being the squared logarithmic error (i.e., $(\log |a_{true}| - \log |a_{predicted}|)^2$ for an entry $a$). Then, the points were binned into 11 bins with equal numbers of points. The figure shows the mean of the errors (second coordinate) of the value pairs with predicted variance (first coordinate) in each of the bins, the color corresponds to the particular algorithm; each group of bars is centered on the minimum value of the associated bin.

## 4.2 Influence of noise level

We generated 10 random mask of size $50 \times 50$ with 200 entries sampled uniformly and a random $(50 \times 50)$ matrix of rank one. The multiplicative noise was chosen entry-wise independent, with variance $\sigma_i = (i - 1)/10$ for each entry. Figure 2(a) compares the Mean Squared Error (MSE) for three algorithms: Nuclear Norm (using the implementation Tomioka et al. [10]), OptSpace [5], and Algorithm 3. It can be seen that on this particular mask, Algorithm 3 is competitive with the other methods and even outperforms them for low noise.

## 4.3 Prediction of estimation errors

The data are the same as in Section 4.2, as are the compared algorithm. Figure 2(b) compares the error of each of the methods with the variance predicted by Algorithm 4 each time the noise level changed. The figure shows that for any of the algorithms, the mean of the actual error increases with the predicted error, showing that the error estimate is useful for a-priori prediction of the actual error - independently of the particular algorithm. Note that by construction of the data this statement holds in particular for entry-wise predictions. Furthermore, in quantitative comparison Algorithm 4 also outperforms the other two in each of the bins. The qualitative reversal between the algorithms in Figures 2(b) (a) and (b) comes from the different error measure and the conditioning on the bins.

# 5 Conclusion

In this paper, we have introduced an algebraic combinatorics based method for reconstructing and denoising single entries of an incomplete and noisy matrix, and for calculating confidence bounds of single entry estimations for arbitrary algorithms. We have evaluated these methods against state-of-the art matrix completion methods. Our method is competitive and yields the first known a priori variance bounds for reconstruction. These bounds coarsely predict the performance of *all* the methods. Furthermore, our method can reconstruct and estimate the error for single entries. It can be restricted to using only a small number of nearby observations and smoothly improves as more information is added, making it attractive for applications on large scale data. These results are an instance of a general algebraic-combinatorial scheme and viewpoint that we argue is crucial for the future understanding and practical treatment of big data.

## Footnotes

[1]In particular, if $A$ is sampled from a continuous density, then the set of non-generic $A$ is a null set.

[2] Any fixed orientation of $K_{m,n}$ will give us the same result.

[3] The zero-variance case corresponds to exact reconstruction, which is handled already by Theorem 2.4.

[4]The multiplicative noise assumption causes the observed entries and the true entries to have the same sign. The change of sign can be modeled by adding another multiplicative binary random variable in the model which takes values $\pm 1$; this adds an independent combinatorial problem for the estimation of the sign which can be done by maximum likelihood. In order to keep the exposition short and easy, we did not include this into the exposition.

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
