[Supplementary Material]

# A   Correctness of Algorithm 2

In this appendix we show:

**Theorem A.1.** *For any graph $G$ and any pair of vertices $s$ and $t$ in the same connected component of $G$, Algorithm 2 computes a basis for the $s$–$t$ path space.*

*Proof:* We adopt the conventions of Section 2, so that $G$ is a bipartite graph with $m$ blue vertices, $n$ red ones, and $e$ edges oriented from blue to red. Recall the isomorphism, observed in the proof of Theorem 2.6 of the $\mathbb{Z}$-group of the polynomials $L_C(\cdot)$ and the oriented cycle space $H_1(G, \mathbb{Z})$.

Let $c$, be the number of connected components of $G$ and define $\beta_1(G) = e - n - m + c$ (the first Betti number of the graph). Some standard facts are that: (i) the rank of $H_1(G, \mathbb{Z})$ is $\beta_1(G)$; (ii) we can obtain a basis for $H_1(G, \mathbb{Z})$ consisting only of simple cycles by picking any spanning forest $F$ of $G$ and then using as basis elements the fundamental cycles $C_e$ of the edges $e \in E \setminus F$. This justifies step 4.

Let $(i, j)$ be an edge of $G$. Define an $i$–$j$ *path* (shortly, just a *path*) to be a subgraph $P$ such that, for generic rank one $A$, $L_P(A) = -x_{(i,j)}$; the set of all $i$–$j$ paths is the $i$–$j$ *path space*. By Theorem 2.6, we can write any path as a $\mathbb{Z}$-linear combination of $x_{(i,j)}$ and oriented cycles. From this, we see that the rank of the path space is $\beta_1(G) + 1$ and the graph theoretic identification of elements in the path space with subgraphs that have even degree at every vertex except $i$ and $j$. Thus, if $(i, j)$ is an edge of $G$, step 5 is justified, completing the proof of correctness in this case.

If $(i, j)$ was not an edge, step 1 guarantees that the dummy copy of $(i, j)$ that we added is not in the spanning tree computed in step 3. Thus, the element $P_{(i,j)} = C_{(i,j)} - x_{(i,j)}$ computed in step 6 is a simple path from $i$ to $j$. The collection of elements generated in step 6 is independent by the same fact in $H_1(G \cup \{(i, j)\}, \mathbb{Z})$ and has rank $\beta_1(G) + 1$. Since none of the elements put a non-zero coefficient on the dummy generator $x_{(i,j)}$, we are done.  □