[Reviews · NeurIPS 2013]

Submitted by Assigned_Reviewer_7

The paper studies the noisy matrix completion problem, focusing on entry-wise error guarantees. Via algebraic connections, they derive an algorithm for computing the minimum variance unbiased estimator for any missing entry. They also provide an explicit expression for the minimum variance, show how these results can be extended to higher rank matrices, and show promising comparisons with state-of-the-art matrix completion algorithms.

The paper is inspired by a recent paper on noiseless matrix completion, which shows that the set of entries recoverable depends only on the mask and in the rank-one case, this dependence is made explicit by the connectivity properties of the bipartite graph induced by the mask. In the noiseless case, one can recover entries by solving a system of polynomial equations defined by paths in the bipartite graph. With the appropriate noise model (they use multiplicatively-centered finite variance noise) this algebraic connection suggests that a polynomial expression from any path between i,j in the mask yields an unbiased estimator of the entry A_{ij}.

Minimizing the variance involves taking the appropriate linear combinations of the polynomials defined by paths between i,j. To do this, the authors first give an explicit expression for the variance depending only on the coefficients, the graph itself, and the the variance of the entries of the matrix (assumed to be known). Minimizing this expression leads to the MVUE.

For higher rank matrices, the polynomial equations aren't given by paths and cycles but by higher-order subgraphs, but the same arguments can be applied.

The simulations conducted show how the variance of individual entries depends on the structure of the mask and also compare their estimation procedure with the nuclear norm minimization and the OptSpace algorithm. In the latter, this approach is favorable in the low noise regime, but their error estimates are useful independently of algorithm choice.

The paper presents a novel analysis of noisy matrix completion and uses nice connections to algebraic combinatorics and graph theory to arrive at interesting results about estimation. The paper is well written and fairly easy to understand, addresses an interesting problem with original techniques. Focusing on the rank-one case limits the direct applicability of the work but at the same time it significantly aids readability and extensions to higher rank settings seem easy enough if one can construct a basis for the relevant "circuits." Another potential weakness is the use of multiplicative noise rather than additive noise, which is much more standard in the statistics community and the existing works on matrix completion.

It would be nice to give frobenius norm error bounds under random sampling as a function of the noise variance. I suspect this is not too challenging in the model specified using results from random graph theory but it would be even better if it could be done with additive noise, which would enable precise comparisons with related matrix completion papers.
Summary: The paper presents novel insights into the matrix completion problem via connections from algebraic combinatorics and graph theory, resulting in a minimum variance unbiased entry-wise estimator. While there are some auxiliary results that would add to the quality of the paper, it is a strong contribution as is.

Submitted by Assigned_Reviewer_8

The matrix completion problem is considered. Unlike other approaches, the current paper does not aim at estimating all entries of the matrix, but rather to understand which entries can be “well” estimated. It also provides an algorithm for the actual estimation. The authors concentrate on the rank-1 case but suggest a way to generalize their ideas to the more general rank-r case. Comparisons with state of the art algorithms are provided on simulated rank-1 examples.

The paper is interesting but quite technical, which makes it difficult to read. Some intuitive explanations or figures would help.

The authors discuss mainly the rank-1 case but comment on the general case as well. Rank-1 matrices are rather simple and well-understood. I think the proposed theory would only be practical if (provably or at least experimentally) confirmed to work on higher rank matrices.

The authors discuss “generic” matrices (for example in Theorem 2.3). Real-world applications however often have non-generic special interpretable properties coming from the particular application.

Currently, the presented experiments are for small (50 x 50) rank-1 simulated matrices. It would be interesting to see how the proposed algorithm behaves on real-world datasets.

Some typos:
- ln. 83: “An matrix”
- ln. 283: “difficulties lies”
- ln. 412: “as are the compared algorithm”
Summary: The paper addresses an important problem and approaches it in a non-standard way. However the presentation is quite technical and while the paper aims to solve the (large-scale) low-rank matrix completion problem, the experiments are limited to rank-1 simulated examples of size 50 x 50.

Submitted by Assigned_Reviewer_9

The paper proposes a framework for reconstructing and denoising rank-1 matrices by topology of graphs that is specific to the problem.

In line 178, the statement "one can show that no estimator for A_{11} has bounded variance" does not seem true. Simply one can use a upper and lower bound for their estimates, for instance, the estimated value must has absolute value less than some constant. This is commonly used for most matrix completion algorithm.

The proof of Theorem B.1 in the appendix is flawed. In line 549, it is true that d'_{i,j,i,j} must be 0 according to the assumption, however it's not clear weather d'_{i,j,i,j, k, l} must also be 0 for $(k,l) \neq (i,j)$ as this parameter does not appear in the expectation of a_{i,j}. The following conclusion that $f$ must be linear to $b_{i,j}$ is not true as it is easy to construct non-linear unbiased estimator. For instance the estimator $\hat{a_{1,1} } = b_{1,1} + b_{1,1}^2 * ( b_{2,2} - b_{2,3} - b{3,2} + b_{3,3}) $ is unbiased as the expectation of ( b_{2,2} - b_{2,3} - b{3,2} + b_{3,3}) = 0 for rank-1 matrices and is independent of b_{1,1}. The proof should be modified.

For multiplicative noise, a direct way to solve the matrix completion problem is by solving the following optimization problem: $min_{u,v} (b_{i,j} - u_i - v_j)^2$ where u_i and v_j corresponding to the log of the rank-1 component vectors. It would be interesting to compare this estimator to the one proposed in the paper.
Summary: The paper provides a new approach of matrix completion by using topology of graphs from the data. The overall idea is novel and may lead to new algorithms for matrix completion with higher rank.
Author Feedback

Author rebuttal: We thank all the reviewers for their helpful feedback.

Below are some point by point responses and clarifications.


Reviewer_7:
>... extensions to higher rank settings seem easy enough if one can construct a basis for the relevant "circuits."

The circuits can be explicitly constructed using algorithms based on linear algebra, see [6].

>Another potential weakness is the use of multiplicative noise rather than additive noise, which is much more standard ...

Additive noise is typical in the literature. It is also typical to restrict to a compact set of true matrices. We do not do this, multiplicative noise suffices to ensure finite variance estimators.

>It would be nice to give frobenius norm error bounds under random sampling as a function of the noise variance. I suspect ... random graph theory but it would ... be even better ... with additive noise,

With multiplicative noise, we need only to combine our technique here with results on cycle counts/structure in random graphs. Changing the noise model is mostly a matter of technical effort.


Reviewer_8
>Unlike other approaches, the current paper does not aim at estimating all entries of the matrix, but rather to understand which entries can be "well" estimated.

This is a major difference between this paper and others in the area.

>The paper is interesting but quite technical, which makes it difficult to read. Some intuitive explanations or figures would help.

The key novel idea is averaging estimators obtained via different polynomial expressions for the same entry. We also tried to indicate how to view things from the perspectives of graph theory, algebra, and optimization.
This is also new.

>The authors discuss "generic" matrices (for example in Theorem 2.3). Real-world applications however often have (close to) low complexity matrices. Are these matrices also generic in the sense of the proposed theory?

This notion of generic is under the condition of rank 1. It implies "with high probability" statements for, e.g., integer matrices.

>The authors discuss mainly the rank-1 case but comment on the general case as well. It remains however unclear ... are they still work in progress, or is it that the algorithm is extendable ...?

The tools presented in the paper and [6] identify efficiently the subgraphs from which we can construct the estimator. While in principle an algorithm is outlined in section 3.3, its final form is still a work in progress.

>Currently, the presented experiments are for small (50 x 50) rank-1 simulated matrices. It would be interesting to see how the proposed algorithm behaves on real-world datasets.

We ran our experiments using un-optimized MATLAB code on a MacBook Air. Even with this setup, complete reconstruction of sparse matrices on the order of 1000 x 1000 are feasible. For one single entry, much larger matrices are feasible. We chose the small example size for validation purposes.


Reviewer_9
>The paper proposes a framework for reconstructing and denoising rank-1 matrices by topology of graphs that is specific to the problem.

Viewing the homology cycles as circuits in a graphic matroid, this part generalizes to the "completion matroids" of [6] for higher rank. For ease of reading, we restricted to the rank 1 case, which leads to more familiar objects (graphs).

>... one can use a upper and lower bound for their estimates, for instance, the estimated value must has absolute value less than some constant. This is commonly ...

Our theory does not require this assumption, and we do not make it. We use multiplicative noise instead.

>The proof of Theorem B.1 in the appendix is flawed.

Thank you very much for pointing this out. Our proof of Theorem 3.10 is indeed wrong, since it doesn't use the hypothesis that the noise is uncorrelated but not necessarily independent. We have sharpened the statement and repaired the proof.
A detailed sketch can be found at
http://pastebin.com/qF67PUw5
(we'll include the complete proof in the final version)